# The Effects of Academic Stress and Upward Comparison on Depression in Nursing Students during COVID-19

**DOI:** 10.3390/healthcare10102091

**Published:** 2022-10-20

**Authors:** Eunju Kwak, Seungmi Park, Ji Woon Ko

**Affiliations:** 1Department of Nursing Science, Chungbuk National University, Cheongju 28644, Chungbuk, Korea; 2Department of Nursing Science, Sunmoon University, Asan-si 31460, Chungnam, Korea

**Keywords:** COVID-19, academic stress, depression, nursing students, social comparison, upward comparison

## Abstract

The coronavirus disease 2019 (COVID-19) pandemic has created a global long-term education crisis, which has negatively affected the psychological well-being of nursing students. This study aims to determine the effect of academic stress and upward comparison on depression among nursing students during the COVID-19 pandemic. A convenience sample of 271 junior and senior nursing students from four universities in South Korea was selected. The SPSS/WIN 28.0 program was employed for the data analysis, and multiple regression analysis was performed to confirm the effect of academic stress and stress from upward comparison on depression. The study results show that the regression model was significant (F = 7.60, *p* < 0.001). Moreover, age over 25 (β = 0.15, *p* = 0.006), academic stress (β = 0.31, *p* < 0.001), and upward comparison (β = 0.18, *p* = 0.002) explained 19.0% of depression among the participants. Developing and testing the effect of programs that address academic stress and upward comparison may be necessary to control depression in nursing students. Furthermore, in response to COVID-19, efforts must be made to include these interventions in the curriculum for nursing students on a consistent basis.

## 1. Introduction

Coronavirus disease 2019 (COVID-19) was identified in Wuhan, China, at the end of December 2019. Since then, COVID-19 has spread rapidly throughout China and globally [1]. Since the first case of COVID-19 in Korea in January 2020, Korea has implemented “social distancing” to minimize human-to-human contact to prevent the spread of infection [2]. This meant that many education facilities, in Korea and worldwide, had to switch from face-to-face education learning to online, remote, or distance learning activities [3].

The COVID-19 pandemic has created a global long-term education crisis, which has negatively affected the psychological well-being of nursing students [4]. According to a meta-analysis of 27 cross-sectional studies of nursing students, the prevalence of depression among nursing students was 34.0% before the COVID-19 pandemic [5]. During the COVID-19 pandemic, the prevalence of depressive symptoms was 31.1% among nursing students in Japan [6]. In a study conducted in Italy, almost half the nursing students who participated in the study had depressive symptomology [7]. Furthermore, the measures of nursing students’ depression tripled when compared to pre-pandemic levels in the US [8].

The educational environment of nursing students has experienced drastic changes as clinical practice in hospitals for nursing students was suspended [8]. Nursing students in Korea undertake 130–140 credits for major and liberal arts classes over four years. Approximately 20% of these credits come from clinical field practicums, which require at least 1000 h of practical experience leading to a tight academic schedule [9]. Additionally, nursing students are under enormous stress managing the heavy workload of a rigorous curriculum, along with the burden of preparing for the national nursing exam [10]. Because nursing education consists of clinical practice, students and educators may be concerned about insufficient clinical skills development during the COVID-19 pandemic. This uncertainty and overcompensation of education due to the lack of clinical education can lead to stress for university nursing students [8]. Excessive academic stress may cause difficulties with understanding the course materials and depression [11], negatively affect physical health [12], and lead to the abandonment of studies [13,14]. It is also a risk factor for suicide and depression among college students [11,15]. Therefore, managing academic stress where necessary and possible is critical.

Upward comparison is the process in which individuals compare themselves with others who they feel are better than themselves [16]. Recently, with an increase in the number of people using social network services (SNSs), “showing off” has become a popular trend. SNSs have become spaces where specific, tangible images that people may envy are expanded and reproduced [17]. Since the COVID-19 pandemic, interpersonal social media and online interactions have become central, and individuals tend to rely heavily on social networks to validate their experiences to support their well-being and/or increase their self-esteem [18]. The COVID-19 pandemic has increased the relationship between social media use and mental health [19]. According to the social comparison theory, individuals evaluate their opinions and abilities by comparing themselves with others. Upward comparison may cause a feeling of relative deprivation, and people who engage in such comparison may experience stress owing to disappointment or frustration with themselves or an increase in negative emotions, such as depression, anger, self-pity, and anxiety [20]. A previous study has reported that upward comparison has a close correlation with depression [21].

The COVID-19 pandemic has brought about significant challenges in the lives of nursing students. In particular, students had to study through non-face-to-face classes such as online or distance learning, which became a new cause of academic stress. Additionally, they began to experience more upward comparison related to online interactions since the COVID-19 pandemic. The present study analyzes how academic stress and upward comparison affect depression in nursing students during the COVID-19 pandemic.

Nursing students are critical to improving the quality of future patient care. This study aims to examine the effects of academic stress and upward comparison on depression among nursing students during the COVID-19 pandemic to prepare basic data for the development of programs that will prevent depression among nursing students and improve their mental health.

## 2. Materials and Methods

### 2.1. Study Design

This descriptive quantitative study aims to investigate the effects of academic stress and upward comparison on depression among nursing students during the COVID-19 pandemic in South Korea.

### 2.2. Participants

The participants of this study were selected by convenience sampling and included students who were in their third and fourth years at the nursing colleges of four universities located in Chungbuk, Chungnam, Daejeon, and Jeonbuk in South Korea. This study is for nursing students, and students’ voluntary participation may not be possible to achieve in the relationship between professors and students when sampling the subjects. Therefore, the sample was biased because the participants’ voluntary participation was not used without using a randomized method. The specific target selection criteria are as follows:Those who are currently enrolled in a nursing college and have experienced online practice due to COVID-19.Those who can take an online survey using a smartphone or PC. This study’s participants are students who entered the third and fourth grades at the time the study was conducted in 2018 and 2019.

They were in their first and second grades at the end of 2019 when COVID-19 began and mainly received online non-face-to-face classes due to the risk of the spread of the infectious disease. Therefore, clinical practice was sometimes replaced by online simulations and conferences, and compared to existing nursing students, they had less clinical practice experience. The minimum sample size was calculated as 254 people using G*Power 3.1.9.7. (Düsseldorf University, Dusseldorf, Germany), a sample number calculation program, through regression analysis with a significance level (⍺) of 0.05, a power of 0.95, a medium effect size of 0.10, and 10 predictors (right general features and two independent variables). Accordingly, data were collected from 271 participants in this study.

### 2.3. Data Collection

Data were collected from 15 to 27 February 2022. After the researcher explained the study’s purpose and procedure, 275 students were asked to participate in an online survey using their Google accounts. The survey was similar to online surveys conducted by the schools through student accounts. It took approximately 10 min to complete. Out of the 275 students asked to participate in the survey, 271 voluntarily participated, and their responses were used for the final data analysis.

### 2.4. Measures

#### 2.4.1. Depression

In this study, depression was measured using the Depression Scale of the Center for Epidemiologic Studies (CES-D) developed by the National Institute of Mental Health (NIMH) in the United States (US) [22]. This scale was adapted into three types of Korean CES-D scales and integrated into the Korean CES-D scale developed by Cheon et al. [23].

The scale consists of 20 items, scored on a 4-point Likert scale ranging from 0 points for “extremely rare” to 3 points for “almost always.” A high total score on this scale indicates a high level of depression among the nursing students. The possible score range is 0 to 60, with a score of 16 or higher indicating mild depression and a score of 25 or higher indicating major clinical depression that requires professional counseling and treatment. The tool’s reliability at the time of its development was a Cronbach’s ⍺ value of 0.91, and in this study, it was 0.84.

#### 2.4.2. Academic Stress

Academic stress was measured using a scale developed by Oh and Cheon [24] and revised by Lee [25] for college students. This scale consists of 42 items scored on a 5-point Likert scale ranging from 1 point for “strongly disagree” to 5 points for “strongly agree.” A high total score on this scale indicates a high level of academic stress among the nursing students. The reliability at the time of development of this tool was a Cronbach’s ⍺ value of 0.93, and in this study, it was 0.93.

#### 2.4.3. Upward Comparison

Upward comparison was measured using the Social Comparison Discrepancy scale developed by Solberg et al. [26]. This scale was revised and supplemented by the eight new comparative dimensions of Yang and Song [27] that measure the realistic experience of individuals in a Facebook-use environment; it was then extended to profile-based SNS by Lim [28]. A high total score on this scale indicates a high tendency for upward comparison among the nursing students. All items are scored on a 7-point Likert scale ranging from 1 point for “strongly disagree” to 7 points for “strongly agree.” The reliability at the time of development of this tool was a Cronbach’s ⍺ value of 0.97, and in this study, it was 0.95.

### 2.5. Data Analysis

The collected data were analyzed using the SPSS/WIN 28.0 program (IBM Corp., Armonk, NY, USA). The general characteristics of the participants were analyzed using frequency analysis and percentage. The variable of depression, academic stress, and upward comparison were analyzed using mean and standard deviation. An independent t-test, one-way analysis of variance, and Scheffé’s post hoc analysis were performed on the differences in depression according to the general characteristics of the participants. To investigate the correlation between depression, academic stress, and upward comparison in participants, the Pearson correlation coefficient was used. Finally, the factors affecting depression among nursing students were analyzed by conducting multiple regression analysis.

### 2.6. Ethical Consideration

Information on the necessity and study purpose, research method, required time, possibility of participants dropping out in the middle of the research, voluntary participation, and benefits and disadvantages of participation were considered before commencing data collection. Information regarding the management and confidentiality of personal information, that is, that participation was voluntary and could be terminated at any time without consequence, that the personal information of the participants would be thoroughly protected, and that the collected data would be used only for this study, were included in the front part of the questionnaire, as well as in an explanatory text describing the contact information of the author. Predetermined rewards were presented to the participants prior to the study to encourage participation. The study received ethical approval from the Chungbuk National University Institutional Review Board of CBNU-202203-HR-0263.

## 3. Results

### 3.1. General Characteristics of Participants

The participants include 239 women (88.2%), with 237 (87.3%) being under the age of 24. Of the students, 112 (41.3%) had grade point averages (GPAs) in the last semester between 3.5 and 4.0, and 177 (65.3%) indicated they were “satisfied” with their major. Regarding type of residence, the numbers of students living with family members and those not residing with their families were similar. The number of participants without a religion was 175 (64.6%), while 96 (35.4%) participants indicated a religious affiliation. In addition, the most common clinical practice period was under one semester, comprising 113 participants (41.7%). The general characteristics of the study participants are shown in Table 1.

### 3.2. Depression, Academic Stress, and Upward Comparison Scores

The average score for depression among the participants was 15.03 (7.66) out of a possible score of 60; the academic stress average score was 2.54 (0.52) out of 5, and the average upward comparison score was 3.42 (1.51) out of 7 (Table 2).

### 3.3. Difference in Depression according to General Characteristics

The depression levels of the participants differed significantly according to age, year of college, satisfaction with major, and clinical practice experience. Participants aged 25 or older showed higher depression levels than those aged 24 or younger. The post hoc analysis indicated that depression levels were higher among those with an average level of satisfaction concerning their major than among those with a high level of satisfaction. Depression levels were also higher among those with two semesters of clinical practice experience than those with four semesters of experience (Table 1).

### 3.4. Correlation between Depression, Academic Stress, and Upward Comparison

Depression had a significant positive correlation with academic stress (r = 0.36, *p* < 0.001) and upward comparison (r = 0.26, *p* < 0.001).

### 3.5. Factors Affecting Depression

Multiple regression analysis was performed on general characteristics including age, GPA, major satisfaction, and clinical practice period. These were confirmed to precede a statistically significant difference in depression as independent variables after processing them as dummy variables. Academic stress and upward comparison showed significant correlation as independent variables. The assumption of regression analysis was tested, and the Durbin–Watson statistic was found to be close to 2, confirming that there was no autocorrelation problem with adjacent error terms. The tolerance limit (TOL) and variance expansion factor (VIF) indicated no multicollinearity of the independent variables. The regression model was significant after multiple regression analysis, with 25 years or older, academic stress, and upward comparison explaining 19.0% of depression among nursing students. Academic stress affected the depression of nursing students the most, and high upward comparison led to high levels of depression among nursing students aged 25 or older (Table 3).

## 4. Discussion

This study aimed to investigate the effects of academic stress and upward comparison on depression among nursing students during the COVID-19 pandemic. The results indicate an average depression score of 15.03 out of 60 points. This value was higher than the average depression score of 12.28 out of 60 points, which was obtained in a study of nursing students conducted in 2019 just before the pandemic [29]. Although both CES-D scores were below the 16-point cut-off score, the depression scores in this study were closer to those classified as depression. Anxiety and depression have been shown to have more than 30% prevalence among nursing students in Japan, a few months after the onset of the COVID-19 pandemic [8]. Santangelo et al.’s [6] study of nursing students in Italy showed that almost half of the participants had depressive symptomology, which was associated with being a woman and a smoker as well as having a low perceived economic status and low perceived health status. Furthermore, during the early months of the pandemic in the US, the measures of nursing students’ depression tripled when compared to pre-pandemic levels [7]. The COVID-19 pandemic was observed to have resulted in a long-term education crisis, which has negatively impacted the psychological well-being of college students, including nursing students [5]. In a previous study, the depression level of nursing students was lower than that of other college students during the COVID-19 pandemic [7]. This could be because the nursing curriculum enables students to foster the social bonds and attachments they need to develop a sense of belonging. The characteristics of the nursing education system may have provided some protection from pandemic-related stress and depression among nursing students [7].

In this study, academic stress was low, with an average of 2.54 out of 5. Although direct comparison was difficult owing to the difference in the scales predominantly used before the pandemic, the score was lower than 2.75 out of 5 [13] and 2.33 out of 4 [30] for nursing students in other studies. Nursing students in several countries expressed stress due to e-learning activities [3]; this is because nursing education cannot all be conducted through e-learning as practical aspects are difficult to learn and convey in these methods. Results from a global survey showed that students reported increased workloads while learning online [31]. Another qualitative study of medical students in Saudi Arabia has shown that online learning was well-received and that the benefits of such education were recognized by students [32]. In earlier studies, the academic stress of students in their first, second, and third years of nursing college was measured. Considering that more than 40% of this study’s participants were in their fourth year, their adaptation to the learning method and curriculum may have contributed to lowering the level of academic stress compared to that of earlier studies. Despite an increased reliance on social networks to maintain self-esteem and validate their experiences following the onset of the pandemic [18], upward comparison in this study showed an average of 3.42 points out of 7, which is a median value between “strongly disagree” and “neutral,” in line with the results of earlier pre-pandemic studies conducted among university students [33,34].

Academic stress and upward comparison were found to be the predictors of depression among nursing students in this study. A study of university students in Lebanon concluded that the drastic changes that occurred during the pandemic began to disrupt students’ learning and create stressful workloads, leading to symptoms of anxiety and depression [32]. The stress levels of first-year students may have increased due to the cessation of education and clinical practice early in their educational careers, lower levels of expertise and skills, greater anxiety about passing classes, and forced impositions [4]. Fourth-grade students’ anxiety regarding graduation and employment has also been identified to affect their stress levels [4,35]. In China, passive social media use was positively associated with upward comparison, which predicted higher levels of stress [36]. In a study of young adults in Romania, upward comparison through social media reduced life satisfaction and increased loneliness during the COVID-19 lockdown [37]. However, the study by Lim [28] conducted with participants who were not university students identified no correlation between upward comparison and depression. Upward comparison appeared to have affected the level of depression more in college students who were not socially independent and were not socially independent and were receiving financial support from their families than in the general population. Furthermore, upward comparison has been found to not only affect depression but also foster a sense of inferiority [16]. Comparisons with others in a better position than oneself lead to emotions such as envy and helplessness, and such negative emotions may further develop into irrational beliefs, interfering with healthy human behavior [38]. In addition, a high tendency for upward comparison may increase the amount of time spent on SNS, even leading to SNS addiction [39]. Therefore, providing counseling to nursing students is necessary to promote healthy behavior and participation in conversion activities to reduce their SNS use. Moreover, nursing students should be encouraged to join mental health intervention programs to assist them in providing adequate nursing care for their patients.

Based on the results of this study, being 25 years or older was identified as an influencing factor for depression. A qualitative study of nursing students in Spain similarly found that switching from face-to-face learning to e-learning during the pandemic was more worrisome for older students than for younger students [32]. In this study, the independent variables, including academic stress and upward comparison, explained approximately 19% of the depression among nursing students, suggesting the need for various future intervention approaches for these factors.

Due to the COVID-19 outbreak, non-face-to-face online classes are increasing unprecedentedly. In this situation, the number of interpersonal relationships among college students is decreasing, and the phenomenon of spending time on social media is increasing. This has negatively impacted their adaptation to college life, leading to increasing loneliness that excessive online use is unable to solve [40]. Global pandemics, such as the 1918 Spanish Flu and the more recent COVID-19, generate significant social attention at first [41]. However, social indifference may become prevalent later through a process of fear. The Spanish Flu was found to stimulate human anxiety and prolonged social isolation, as well as increase depression in those who experienced the pandemic, lowering their overall social achievement [41]. Therefore, nursing students, during the COVID-19 pandemic, may experience a lack of social achievement and increased anxiety and depression after the pandemic is over. As these negative social and psychological experiences can negatively affect new nurses’ workplace adjustment and patient outcomes, consistent attention and intervention should be considered in the future.

This study is significant as it investigates the effects of academic stress and upward comparison on depression among nursing students during the COVID-19 pandemic and prepared basic data to explain the necessity of developing mental health nursing intervention programs. To prevent depression and reduce academic stress and upward comparison in times of non-face-to-face routines, developing an individual mental health nursing intervention program and follow-ups is necessary, including continuous interventions and observations. However, a limitation exists in the generalizability of this study’s results, as the participants, who were in their third and fourth years in the nursing colleges of four domestic universities, were recruited by convenience sampling.

## 5. Conclusions

This study’s results reveal that academic stress and upward comparison significantly affect depression among nursing students. Therefore, reducing academic stress and upward comparison is expected to help prevent depression. Developing and applying mental health intervention programs for student nurses targeting academic stress and upward comparison, including continuous follow-ups, and verifying their effects on depression are necessary. Furthermore, in response to the COVID-19 imperatives for remote learning, efforts must be made to include these interventions in the curriculum of nursing students on a consistent basis.

## Figures and Tables

**Table 1 healthcare-10-02091-t001:** General Characteristics of Participants and Differences in Depression According to General Characteristics (*n* = 271).

Characteristics	Categories	*n* (%)	Depression		
			M (SD)	t or F(*p*)	Sheffé
Sex	Male	32 (11.8)	13.44 (6.59)	−1.25 (0.211)	
	Female	239 (88.2)	15.24 (7.77)		
Age (years)	≥24	237 (87.5)	14.68 (7.53)	−1.98 (0.049) *	
	≤25	34 (12.5)	17.44 (8.18)		
School year	Junior	152 (56.1)	15.91 (8.10)	2.15 (0.033) *	
	Senior	119 (43.9)	13.91 (6.92)		
Last semester GPA	<3.0 (B0) ^†^	18 (6.6)	16.39 (5.25)	1.21 (0.308)	n/a
(out of 4.5)	3.0 (B0)–<3.5 (B+) ^†^	78 (28.8)	16.18 (8.21)		
	3.5 (B+)–<4.0 (A0) ^†^	112 (41.3)	14.33 (7.28)		
	≥4.0 (A0) ^†^	63 (23.2)	15.03 (7.66)		
Major satisfaction	Satisfied ^a^	177 (65.3)	13.92 (6.79)	5.89 (0.003) *	b > a
	Average ^b^	86 (31.7)	17.31 (8.42)		
	Unsatisfied ^c^	8 (3.6)	15.00 (12.04)		
Living with	Family members	140 (51.7)	15.05 (8.04)	0.06 (0.951)	
	Non-family members or alone	131 (48.3)	15.00 (7.25)		
Clinical practice period	None or 1 ^a^	113 (41.7)	15.27 (7.10)	3.91 (0.009) *	b > d
(semester)	2 ^b^	74 (27.3)	16.97 (9.25)		
	3 ^c^	50 (18.5)	13.48 (6.10)		
	4 ^d^	34 (12.5)	12.26 (6.61)		
Religion	Yes	96 (35.4)	14.52 (7.74)	−0.81 (0.419)	
	No	175 (64.6)	15.31 (7.62)		

M = Mean; SD = Standard deviation; GPA = Grade point average. * *p* < 0.005. ^†^ Academic performance level; A0 = 4.0/4.5, B0 = 3.0/4.5, B+ = 3.5/4.5; a,b,c,d = comparison groups of Scheffe test.

**Table 2 healthcare-10-02091-t002:** Depression, Academic Stress, and Upward Comparison (*n* = 271).

Variables (Total or Scale)	M (SD)	Min	Max
Depression (0–60)	15.03 (7.66)	0.00	49.00
Academic stress (1–5)	2.54 (0.52)	1.02	4.14
Upward comparison (1–7)	3.42 (1.51)	1.00	7.00

M = Mean; SD = Standard deviation; Min = Minimum; Max = Maximum.

**Table 3 healthcare-10-02091-t003:** Factors Influencing Depression (*n* = 271).

Independent Variables		B	SE	β	t (*p*)	TOL	VIF
(constant)		−0.96	2.15		−0.45 (0.656)		
Age	≥25	0.18	0.06	0.15	2.77 (0.006)	0.98	1.02
School year	Junior	0.01	0.06	0.01	1.12 (0.908)	0.50	2.00
Major satisfaction	Satisfied	−0.02	0.13	−0.03	−0.17 (0.863)	0.12	8.41
	Average	0.05	0.13	0.06	0.35 (0.724)	0.12	8.25
Clinical practice	None or 1	0.02	0.09	0.03	0.22 (0.826)	0.24	4.12
Period †	2	0.13	0.80	0.15	1.66 (0.099)	0.36	2.77
(semester)	3	−0.01	0.80	−0.01	−0.14 (0.886)	0.48	2.08
Academic stress		4.58	0.84	0.31	5.44 (<0.001)	0.92	1.09
Upward comparison		0.93	0.29	0.18	3.15 (0.002)	0.91	1.10
Durbin–Watson = 1.929, R^2^ = 0.202, Adj R^2^ = 0.190, F = 7.60, *p* < 0.001							

SE = Standard error; VIF = Variance expansion factor; TOL = Tolerance limit. † Dummy variables.

## Data Availability

The data presented in this study are available on request from the corresponding author. The data are not publicly available due to privacy.

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
