# Peer review of "The Effects of Academic Stress and Upward Comparison on Depression in Nursing Students during COVID-19"

_healthcare, 2022, doi:10.3390/healthcare10102091_

Round 1
Reviewer 1 Report
This paper reports on the results of a questionnaire of nursing students in South Korea in early 2022, investigating academic stress, upward comparison and depression following the Covid-19 pandemic.
This is an important area and the findings would be an important contribution to the evidence of the effects of the pandemic on students' experiences, in particular those working in health-related environments.
I see few problems with the methodology and how the research has been carried out and reported. It seems to be presented very well. I noticed a few times when more detail could be provided and some minor proofreading errors.
Minor editing suggestions:
Intro line 25 - "The COVID-19 was identified" - word missing, The COVID-19 virus?
Line 43 - "Excessive academic stress may cause learning difficulties" - suggest change of wording as "learning difficulties" can be understood as a specific term to note a long term condition such as dyslexia. Perhaps "difficulties with course materials".
Methods line 85 - there is mention of rewards being provided, what were the rewards? 271 of 275 is a very high return rate. If this was a compulsory questionnaire, e.g. for course credit, then that should be noted.
Results lines 140-141 - mention of grade point averages between 3.5 and 4.0 - what does this mean? are they good results? the table later on mentions that this is out of 4.5, but how does these participants' average grade compare with the general average, or with pre-pandemic scores, and what is the pass grade?
Discussion line 192 - I think it's worth reiterating that a higher score indicates more severe symptoms on the depression scale.
Line 264 - the first sentence of this paragraph does not quite make grammatical sense. Consider rephrasing and perhaps splitting into two sentences.
Reviewer 2 Report
Thanks for the opportunity to review this manuscript.
The following comments may help the authors to improve the standard of the manuscript.
1. Need more justification for the rationale of the study.
2. Need justification for the sample selection procedure. Are the scales validated?
3. Need more construct reliability and validity evidences.
Reviewer 3 Report
The manuscript The Effects of Academic Stress and Upward Comparison on Depression in Nursing Students during COVID-19 needs some revision:
1- The objective of of the study and the introduction show that depression is a great concern in nursing students, but there is no evidence that shows the prevalence of depression in this population. Upward comparison seems to be somehow an unrelated variable to academic stress, the reasons for compounding these variables need more explanation.
2- Manuscript needs English editing.
3- 275 students, out of how many? Why random methods have not been used for sampling? How the participants have been selected? Please provide more information about the sampling.
4- Please provide more information about the clinical education of the subjects. Since they were in 3th and 4th years of education, they have been in clinical courses during the covid crisis or not? Please provide more information about their academic and clinical education.
5- Does the depression scale has a cut-off point to show the high possibility of depression or not?
6- I think the tables 1 and 3 can be merged.
7- The discussion is based on the information that are not presented in the result section. For example it says the there was no significant change in depression of nursing students in Covid. but I van not see the status before Covid. Or there are no information about social bonds or adaptation.
Round 2
Reviewer 2 Report
Dear authors
This manuscript is an interesting article in the field of Nursing and Mental Health, after working on the questions suggested by the reviewers, I believe that the article has improved considerably so that it can be published.
Regards,
Author Response
Dear Reviewer 2
We did our best to revise the paper according to reviewers suggestions.
Thank you for your positive comments on the revised paper.
Best regards,